# NnABI4-Mediated ABA Regulation of Starch Biosynthesis in Lotus (*Nelumbo nucifera* Gaertn)

**DOI:** 10.3390/ijms222413506

**Published:** 2021-12-16

**Authors:** Peng Wu, Ailian Liu, Yongyan Zhang, Kai Feng, Shuping Zhao, Liangjun Li

**Affiliations:** 1School of Horticulture and Plant Protection, Yangzhou University, Wenhui East Road No. 48, Yangzhou 225009, China; wupeng@yzu.edu.cn (P.W.); liuailianyz@163.com (A.L.); zhangyongyanyz@163.com (Y.Z.); fengkai@yzu.edu.cn (K.F.); zhaoshuping@yzu.edu.cn (S.Z.); 2Joint International Research Laboratory of Agriculture and Agri-Product Safety of Ministry of Education of China, Yangzhou University, Yangzhou 225009, China

**Keywords:** plant starch, abscisic acid, NnABI4, lotus (*Nelumbo nucifera* Gaertn)

## Abstract

Starch is an important component in lotus. ABA is an important plant hormone, which plays a very crucial role in regulating plant starch synthesis. Using ‘MRH’ as experimental materials, the leaves were sprayed with exogenous ABA before the rhizome expansion. The results showed that stomatal conductance and transpiration rate decreased while net photosynthetic rate increased. The total starch content of the underground rhizome of lotus increased significantly. Meanwhile, qPCR results showed that the relative expression levels of *NnSS1*, *NnSBE1* and *NnABI4* were all upregulated after ABA treatment. Then, yeast one-hybrid and dual luciferase assay suggested that NnABI4 protein can promote the expression of *NnSS1* by directly binding to its promoter. In addition, subcellular localization results showed that NnABI4 encodes a nuclear protein, and NnSS1 protein was located in the chloroplast. Finally, these results indicate that ABA induced the upregulated expression of *NnABI4*, and NnABI4 promoted the expression of *NnSS1* and thus enhanced starch accumulation in lotus rhizomes. This will provide a theoretical basis for studying the molecular mechanism of ABA regulating starch synthesis in plant.

## 1. Introduction

Starch is the main storage form of carbohydrate in higher plants and occupies an important position in the human diet [1]. Starch is also an important factor that determines the edible quality of most crops [2], such as cassava [3], rice [4,5,6] and maize [7]. According to the different molecular structure, starch mainly includes two glucose polymers: amylose and amylopectin [8]. The biosynthesis of plant starch is a complex metabolic process, which requires the synergy of multiple enzymes, including ADP glucose pyrophosphorylase (AGPase), granule bound starch synthase (GBSS), soluble starch synthase (SSS) and starch branching enzyme (SBE) [9,10]. AGPase catalyzes glucose 1-phosphate and ATP to synthesize ADPG, the precursor substance of starch synthesis [11]. GBSS is mainly responsible for the synthesis of amylose and the synthesis of extra-long unit chains (ELCs) in amylose [12]. SSS and SBE are mainly involved in amylose synthesis. Among them, SSS includes four isoforms of SSI, SSII, SSIII and SSIV. SSSII and SSSIII are mainly involved in the synthesis of longer chains [13], while SSSIV may be involved in the formation of starch granules [14]. SSI is mainly responsible for the synthesis of short glucans with a degree of polymerization (DP) of 6–15 [15,16,17]. It has been reported that the activity accounts for about 70% of the total SSS activity in grain endosperm [18]. In sweet potatoes, overexpression of *IbSSI* can cause changes in starch content, composition, particle size and structure [19]. Furthermore, SBE can branch starch chain and contribute to the synthesis of amylopectin [20]. Therefore, exploring the biosynthesis mechanism of plant starch will be of great significance to improve crop quality and yield.

Many studies have shown that ABA is positively correlated with starch synthesis. It was found that the synthesis and accumulation of starch are continuously enhanced with the increase in ABA content in potatoes [21]. The ABA content in the seeds of wheat and barley at the filling stage is positively correlated with the accumulation rate of starch in the seeds [22]. In rice, strong grains with high ABA content had high expression of starch synthesis-related genes [23]. After exogenous ABA treatment, the content of total starch, amylose and amylopectin in mature wheat increased [24]. Spraying exogenous ABA during the flowering stage of winter wheat significantly increases the starch accumulation rate and starch content [25]. Treatment of grape buds with exogenous ABA increases starch synthesis and upregulates the expression of *VvSS1* and *VvSS3* [26]. The molecular mechanism of ABA regulating plant starch biosynthesis has also been studied in recent years. ABI4 in *Arabidopsis* can regulate the expression of starch synthesis-related genes (*APL3* and *SBE2*) and starch degradation genes (*SEX1* and *BMY8*/*BAM3*), thereby affecting the synthesis and accumulation of starch [27]. In maize, ABI4 was found to bind to the CACCG motif of *ZmSSI* promoter to enhance the expression of *ZmSSI* gene [28,29].

Lotus (*Nelumbo nucifera* Gaertn) native to China and India, is a characteristic aquatic vegetable with the largest cultivated area in China [30,31]. Fresh lotus and its processed products are not only durable in storage and transportation, but also rich in nutrients such as starch, protein and vitamins, and are deeply loved by consumers in Japan, Southeast Asia, Europe and the United States [32,33]. The enlarged rhizome is the main edible part of lotus, and the starch content in the rhizome accounts for 10–20% of the total fresh weight [34]. Thus starch synthesis and accumulation in rhizome is an important factor to determine the yield and quality of lotus [32]. In the lotus, it was found that the activity of SS reached the peak in the middle and early period of rhizome expansion, and was significantly positively correlated with the total starch content [35]. In addition, the *AGPase* gene and *GBSS* gene of lotus were identified and analyzed [36,37]. However, research on ABA regulating starch synthesis in lotus has not been reported.

In this study, exogenous ABA was sprayed on lotus leaves before rhizome expansion. Firstly, photosynthetic characteristics of the leaves were measured, and then the relative expression levels of starch synthesis-related genes and ABA pathway genes in the rhizome were determined by qPCR. Further, Y1H and dual luciferase assay were performed to verify the binding and regulation of ABA signal transduction-related genes and key genes of lotus starch synthesis. This will provide a theoretical basis for further analysis of ABA regulation of the starch synthesis network in lotus.

## 2. Results

### 2.1. Effects of Exogenous ABA on Photosynthetic Characteristics of Lotus Leaves

The photosynthetic characteristics of lotus leaves treated with ABA were determined. First, the stomatal conductance was higher than that of CK at 1 and 2 days after ABA treatment, while it was lower than CK at 3, 4, 5 and 6 days after ABA treatment (Figure 1A, Appendix A). The leaf transpiration rates were observed lower in ABA-treated samples than CK after two to six days after ABA treatment. (Figure 1B, Appendix A). Moreover, after ABA treatment, the net photosynthetic rate of leaves was higher than that of the control treatment (Figure 1C, Appendix A). Especially after 4 days of ABA treatment, the net photosynthetic rate (24.65 μmol·m^−2^·s^−1^) increased by 0.84 μmol·m^−2^·s^−1^ compared with CK (23.81 μmol·m^−2^·s^−1^), an increase of 3.5%. These results indicated that exogenous ABA treatment may have a certain impact on photosynthesis.

### 2.2. Effect of Exogenous ABA Treatment on the Total Starch Content of Lotus

To explore whether exogenous ABA treatment has an effect on the synthesis of starch in rhizome of lotus, the content of starch in rhizome at different developmental stages was determined. The total starch content in the lotus rhizomes of the four developmental stages was significantly higher than that of CK after ABA treatment (Figure 2, Appendix A). Especially in the beginning stage of expansion, the total starch content was 1.29 times higher in the ABA treated than that of the control. The results indicated that ABA could promote starch synthesis in rhizome of lotus, and exogenous abscisic acid was an effective method to increase the starch content of lotus during the expansion of rhizome.

### 2.3. Expression Profile of Key Genes in Starch Pathway after ABA Treatment

In order to further explore the molecular mechanism of ABA regulating starch biosynthesis in lotus, qRT-PCR was used to detect the relative expression levels of key genes in starch synthesis after ABA treatment. The relative expression levels of *NnSS1*, *NnSS2*, *NnSS3*, *NnSS4*, *NnGBSS1* and *NnSBE1* in the rhizomes after ABA treatment were measured. After ABA treatment, the expression of *NnSS1* was upregulated and significantly higher than that of the control at the beginning and early stage of rhizome development (Figure 3, Appendix A). *NnSS2* was significantly reduced, except at the beginning stage of rhizome development, which was significantly higher than that of the control treatment (Figure 3, Appendix A). In addition, the expression levels of *NnSS3* and *NnSS4* were downregulated after ABA treatment (Figure 3, Appendix A). Compared with the control, the expression of *NnGBSS1* in the rhizome was downregulated during the swelling process (Figure 3, Appendix A). Moreover, the expression level of *NnSBE1* was significantly upregulated at the beginning and early stage of expansion in the rhizomes (Figure 3, Appendix A). The above results found that only *SS1* and *SBE1* expression were upregulated during the four stages of rhizome development after ABA treatment, indicating that ABA may promote rhizome starch synthesis by enhancing *NnSS1* and *NnSBE1* expression.

### 2.4. Expression Profile of Key Genes in ABA Signaling Pathway after ABA Treatment

The relative expression levels of ABA signal transduction key genes such as *ABI3*, *ABI4* and *ABI5* after ABA treatment were determined. *ABI3* was upregulated during the beginning and early stages of rhizome development, and was significantly downregulated during the middle and late stages of rhizome development (Figure 4, Appendix A). The expression of *ABI4* was significantly higher than that of the control at all four stages of rhizome development (Figure 4, Appendix A). *ABI5* was higher than the control only in the early stage of rhizome development, and its expression was significantly downregulated in the other three stages (Figure 4, Appendix A). These results indicate that *ABI4* is likely to be a key gene for ABA to regulate starch biosynthesis in lotus rhizomes. It has been reported that *ZmABI4* can bind to the promoter of *ZmSS1* to enhance the expression of *ZmSS1*, thereby improving starch biosynthesis in maize [28]. Therefore, it is speculated that ABI4 in lotus may play a key role in ABA regulating starch biosynthesis.

### 2.5. Sequence Analysis of ABI4

The conserved motifs of ABI4 of these species were analyzed by MEME online software. As shown in Figure 5A, these seven species contain five identical motifs, which are motif1, motif2, motif3, motif4 and motif5. Motif8 was found in all species except *Amborella trichopoda*, *Arabidopsis thaliana*, *Brassica rapa*, *Populus trichocarpa* and *Nelumbo nucifera*, which all contain motif7. Furthermore, *A.*
*thaliana* and *B.*
*rapa* contain motif6 and motif10, and *Carica papaya* and *Vitis vinifera* contain motif9 (Figure 5A). Meanwhile, analysis of the ABI4 protein domains of seven species found that they all contain the conserved domain AP2, all of which are 52 amino acid residues in length (Figure 5A). Sequence alignment shows that within the domain, the 33rd and 37th amino acid residues of *C.*
*papaya* and *A.*
*trichopoda* are S and E, respectively, while the others are A and D. The 47th and 48th amino acid residues were not conserved in the seven species. The 47th position of lotus is L, which is the same as *C.*
*papaya*, and the 48th position is V (Figure 5B). In addition, gene structure analysis showed that the *ABI4* genes of these seven species have only exons, but no introns (Figure 5C).

### 2.6. NnAB14 Directly Promotes the Expression of NnSS1

In order to verify whether NnABI4 can directly regulate the expression of *NnSS1*, a yeast one-hybrid experiment was conducted. As shown in Figure 6A, NnABI4 can bind to the promoter of *NnSS1*, which indicates that ABI4 can directly regulate the expression of *NnSS1*. Dual luciferase assay was used to explore the regulatory effect of NnABI4 on *NnSS1*. The results show that the relative luciferase level increased by 2.67 times after co-expression of NnABI4 and *NnSS1* in tobacco leaves (Figure 6B, Appendix A). Y1H and dual luciferase assay confirmed that NnABI4 could bind *NnSS1* promoter and promote *NnSS1* expression in lotus. Therefore, it can be concluded that ABA upregulates the expression of *NnABI4*, and NnABI4 promotes the expression of *NnSS1* and ultimately promotes the synthesis of starch.

In addition, subcellular localization analysis of NnABI4 and NnSS1 proteins was performed, and the plasmids of *35S:ABI4-GFP*, *35S:SS1-GFP* and *35S:GFP* were transiently overexpressed in Arabidopsis protoplasts. The green fluorescence signal of *35S:SS1-GFP* was detected in the chloroplast, and *35S-GFP* control was detected in the nucleus and cytoplasm (Figure 6C). Moreover, the green fluorescent signal of *35S:ABI4-GFP* is detected in the nucleus (Figure 6C), which indicates that the ABI4 protein is a nuclear protein.

## 3. Discussion

Photosynthetic products are the basis of plant starch biosynthesis, and the strength of photosynthesis has an important influence on the synthesis of starch [38]. Therefore, the photosynthetic characteristics of the leaves after ABA treatment were measured. The stomatal conductance and transpiration rate of lotus leaves decreased, and the net photosynthetic rate increased. This indicates that ABA has an effect on the photosynthesis of leaves, but its specific mechanism still needs to be further explored. In our study, starch accumulation in the rhizomes of lotus at all four developmental stages was significantly increased after ABA treatment. Phytohormones are involved in the transportation and distribution of photochemical compounds from the source organ (leaf) to the sink organ (fruit), as well as the accumulation and metabolism in the fruit [39]. Exogenous ABA treatment could increase sugar accumulation in peach fruit [40]. ABA can promote the transportation of photosynthetic products from leaves to fruits in apples [41,42], and can promote the transfer of assimilates from stems to grains in rice and wheat [43,44]. Therefore, ABA treatment induced the synthesis of more starch from lotus rhizomes, indicating that ABA may promote the transport of assimilates to lotus rhizomes and provide more raw materials for starch synthesis. This still needs further research.

ABA can regulate the expression of genes in the starch synthesis pathway. In this study, NnSS1, and NnSBE1 genes are all upregulated by ABA induction, which was consistent with the increase in starch content. ABA and sucrose treatment induced OsAPL3 expression and starch biosynthesis in rice suspension cells [45,46]. In maize endosperm, ABA treatment can promote the expression of ZmSS1, while the cotreatment of ABA and sucrose can induce the expression of GBSS1 and SSIIb [29,47,48]. This evidence indicates that ABA may promote starch biosynthesis in lotus rhizomes by promoting the expression of NnSS1 and NnSBE1. In addition, it has been reported that ABA can induce the expression of ABI4 in Arabidopsis [49]. ABI4 plays an important regulatory role in starch synthesis in maize [28]. The expression of NnABI4 in the rhizomes of lotus at four developmental stages was significantly upregulated. Therefore, NnABI4 may serve as a bridge between ABA signaling and starch synthesis in lotus.

ABI4 belongs to the AP2/ERF family of transcription factors and plays an important role in ABA signal transduction [50,51]. Moreover, AB14 can bind to the CE1 element of gene promoters to mediate carbohydrate and ABA-induced target gene expression [52]. ABI4 protein can also bind to CE1-like motif-derived motifs, such as S-box (CACYKSCA) [53] and CCAC motifs [54,55]. In our study, the promoter of NnSS1 in lotus contains CACCG, a CE1 coupling element. Y1H assay and dual luciferase assay confirmed that NnABI4 can bind to the promoter of NnSS1 and promote the expression of NnSS1. These results suggest that *NnAB14* plays a key role in regulating lotus starch synthesis. *ABI4* has also been found in other crops to promote starch synthesis [27,28,29,48], so it may be a common regulatory network in plants. In this study, the expression of SBE1 was also upregulated after ABA treatment, but whether it could be regulated by ABI4 remains to be further explored. In addition, a report found that sucrose and ABA in maize can regulate starch synthesis through ZmEREB156 [56]. This indicates that the network of ABA regulating plant starch synthesis is complex, and further research is needed.

## 4. Materials and Methods

### 4.1. Plant Materials

The main plant varieties ‘MRH’ were planted in the aquatic vegetable experiment base of Yangzhou University, under normal management.

### 4.2. Sample Preparation

When the underground rhizome was not expanded, 5 mg/L ABA was sprayed on the leaves of lotus twice every 10 days, on the first and fourth days, respectively, with a cycle of 30 days. Spraying was applied a total of six times, and each spraying was applied until spraying droplets fell on the leaf surface. Due to the smooth surface of lotus leaves, 0.25% tween 20 was added in both control and treatment groups. Three biological replicates were set up in each treatment. Samples were taken at the beginning (12 July 2020), early (25 July 2020), middle (14 August 2020) and late (5 September 2020) stage of expansion of rhizome of lotus. Three rhizomes with similar size were taken at each stage. They were immediately washed, dried, cut into pieces, put in liquid nitrogen and put in refrigerator at −80 °C for reserve.

### 4.3. Determination of Photosynthetic Parameters of Leaves

The photosynthetic parameters of leaves were measured for six consecutive days on the second day after the first spray of exogenous ABA. The portable photosynthesis measurement system LI-6400XTP can simultaneously measure the leaf photosynthetic rate (Pn), stomatal conductance, and transpiration rate (Ts).

### 4.4. Determination of Starch Content

The lotus rhizomes were dried fresh, powdered and sieved. The total starch content of 0.1 g sample was determined by iodine colorimetry. Starch content (%) = R/[(0.1 × 0.01 × 0.05 × 106)] × 100, R value is the concentration calculated from the standard curve (mg/kg); each sample is set for three biological replicates.

### 4.5. Quantitative Real-Time PCR Analysis

Plant RNA extraction kit (Takara, Dalian, China) was used to extract total RNA from rhizome of lotus at different developmental stages (beginning stage (200 ng/mL), early stage (300 ng/mL), middle stage (280 ng/mL) and late stage (245 ng/mL) of expansion). Then, HiScript^®^IIl RT SuperMixfor qPCR (Vazyme, NanJin, China) was used to reverse transcription into cDNA. The key structural genes of starch synthesis pathway in lotus were analyzed by quantitative reverse transcription-polymerase chain reaction (qRT-PCR). The qRT-PCR reaction was 20 μL, including 10 μL 2 × ChamQ SYBR qPCR Master Mix (Vazyme, Nanjing, China), 0.8 μL positive and negative primer mixture, 1.0 μL cDNA template and 8.2 μL ddH2O, respectively. Primer Premier 5.0 was used for primer design. See Appendix A for the gene specific primer sequences. The *β-Actin* (*NnTUA*, GeneID: 104597659) gene was used as an internal gene expression control: the gene was amplified with forward primer 5′-ACCGCCTCGTCTCTCTTTGG-3′ and reverse primer 5′-CGACCTGAATCCCCGCTTGT-3′. Amplification was performed on the CFX-96 Real-time PCR system (Bio-Rad) using the following real-time fluorescent quantitative PCR program: 95 °C for 30 s, then 95 °C for 10 s and 60 °C for 30 s for a total of 40 cycles. The relative gene expression was calculated by 2^−ΔΔCT^ [57]. Three replicates were performed for each amplification reaction.

### 4.6. Y1H Assays

NnSS1 promoter with a length of 1500 bp was recombined into the pAbAi vector as bait, and the full-length transcription factor NnABI4 cDNA was recombined into the pGADT7 vector as prey. After inhibiting the self-activation of the NnSS1 promoter with Aureobasidin A, the prey ABI4 was transferred to the Y1HGold yeast containing the NnSSI promoter. Subsequently, the bacterial liquid was spread on the SD/−Leu+AbA^200^ plate and allowed to stand and cultivate for 3–6 days.

### 4.7. Transient Dual-Luciferase Detection

To further confirm the binding activity of NnABI4 protein and NnSS1 promoter, dual luciferase assay was performed. First of all, the recombinant plasmids NnABI4-pGreenII 62-SK and proNnSS1-pGreenII 0800-LUC were transferred into *A. tumefaciens* GV3101 strain. Then, the bacteria were resuspended with the infection solution (100 mM Acetosyringone, 0.5 M MES (PH5.6) and 10 mM MgCl_2_) until the OD600 value was 1.0, and the bacterial solution containing NnABI4-pGreenII 62-SK and the bacterial solution proNnSS1-pGreenII 0800-LUC were mixed in a ratio of 1:1, and then left to stand for 3 h, and then the tobacco (*Nicotiana benthamiana*) was injected. According to the fluorescence value measured by the double reporting system, the fluorescence value of the target gene plasmid/the fluorescence value of the internal reference plasmid (i.e., F/R value) was calculated, and the ratio of the target gene plasmid to the control group was calculated, the standard error was calculated and the histogram was made.

### 4.8. Subcellular Localization

NnABI4 and NnSS1 were inserted into the 16318-GFP vector, respectively. Healthy and well-growing leaves of Arabidopsis thaliana were selected and cut into fine wires 0.5–1 mm wide in the environment of 0.4 M Mannitol, and enzymatic hydrolysis was conducted on a 40 rpm shaking table for 2–3 h in the dark. Then, W5 solution (100 mL W5 solution contains: 10 mL 154 mM NaCl, 12.5 mL 125 mM CaCl_2_, 2.5 mL 5 mM KCl, 2 mL 2 mM MES (pH 5.7) and 73 mL ddH2O) of equal volume was added to stop the enzymatic hydrolysis reaction, and the enzymatic hydrolysate was filtered into a centrifuge tube with a round bottom through 40-mesh NYLON MESH, and centrifuged at 100× *g* for 8 min at 4 °C After supernatant was discarded, protoplasts were resuspended with W5 solution, and then resuspended with MMG solution (10 mL MMG solution contains: 4 mL 0.2 M Mannitol, 300 µL 15 mM MgCl_2_, 400 µL 4 mM MES (pH 5.7), and 5.3 mL ddH2O.) after centrifugation. Then, the constructed plasmids of NnABI4-*16318-GFP*, NnSS1*-16318-GFP* and *16318-GFP* were transferred into the 200 uL extracted protoplasts. Finally, the fluorescence was observed under a confocal laser microscope.

## 5. Conclusions

In conclusion, our study showed that exogenous ABA treatment significantly increased the total starch content in the rhizome of lotus. Moreover, the molecular mechanism of ABA regulation of starch synthesis in lotus root was revealed: exogenous ABA increased the expression of NnABI4, which binds to the promoter of NnSS1 and promotes the expression of NnSS1, thereby increasing the synthesis and accumulation of starch (Figure 7). This preliminarily explores the regulation mechanism of ABA in starch synthesis during the expansion stage of lotus rhizome, and also provides new ideas for studying the molecular mechanism of ABA regulation of starch synthesis in plants.

## Figures and Tables

**Figure 1 ijms-22-13506-f001:**
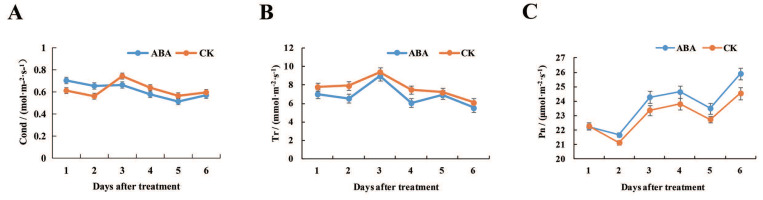
The effects of ABA on photosynthetic characteristics in the leaves. (**A**) Stomatal conductance; (**B**) transpiration rate; (**C**) net photosynthetic rate. Error bars show SD from three biological replicates.

**Figure 2 ijms-22-13506-f002:**
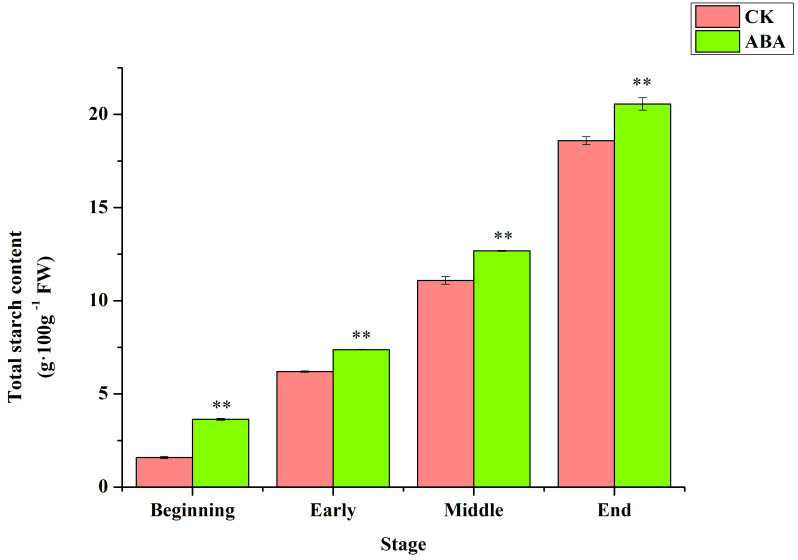
The effects of ABA on total starch content in rhizome of lotus during expansion. The ‘**’ above the histogram indicated the statistical significance at the level of 0.01 (*p* < 0.01). Error bars show SD from three biological replicates.

**Figure 3 ijms-22-13506-f003:**
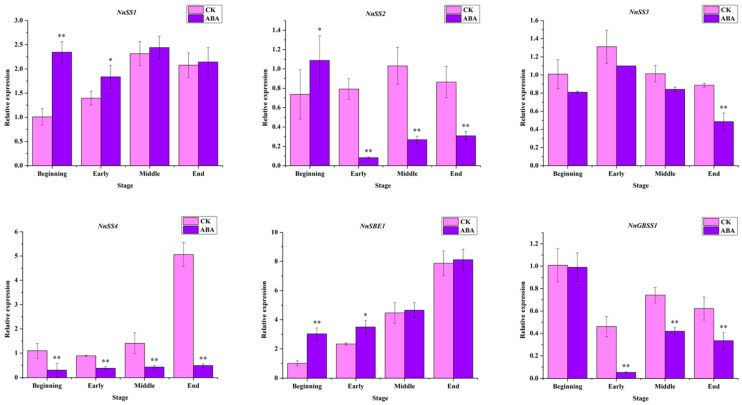
The relative expression of key genes of starch pathway after exogenous ABA treatment. All the data were calculated with three biological repeats. The ‘*’ or ‘**’ above the histogram indicated the statistical significance at the level of 0.05 or 0.01 (*p* < 0.05; *p* < 0.01). Error bars show SD from three biological replicates.

**Figure 4 ijms-22-13506-f004:**
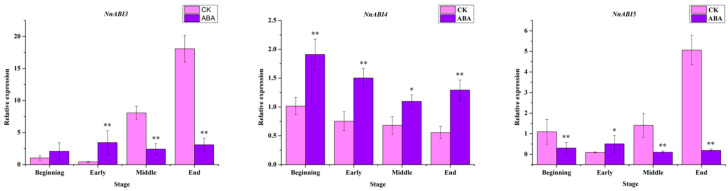
The relative expression of key genes of ABA signal transduction pathway after exogenous ABA treatment. All the data were calculated with three biological repeats. The ‘*’ or ‘**’ above the histogram indicated the statistical significance at the level of 0.05 or 0.01 (*p* < 0.05; *p* < 0.01). Error bars show SD from three biological replicates.

**Figure 5 ijms-22-13506-f005:**
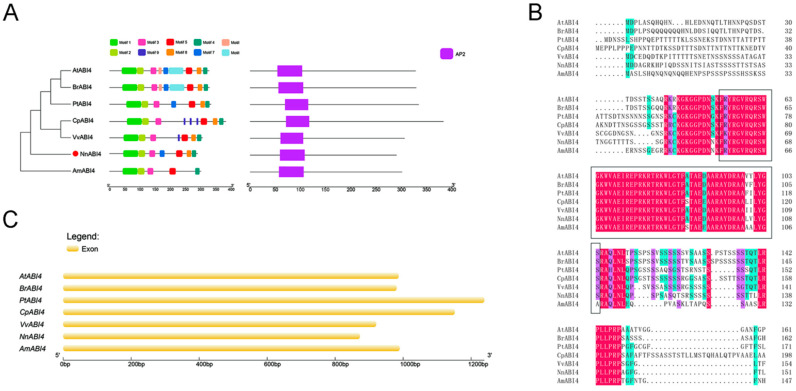
Sequence analysis of ABI4. (**A**) Conserved motifs and domains of ABI4; (**B**) multiple alignment of ABI4 amino acid sequences in different plants. The black box is the sequence of the conserved domain AP2; (**C**) structural analysis of *ABI4*. At (*Arabidopsis thaliana*); Br (*Brassica rapa*); Pt (*Populus trichocarpa*); Cp (*Carica papaya*); Vv (*Vitis vinifera*); Nn (*Nelumbo nucifera*); Am (*Amborella trichopoda*).

**Figure 6 ijms-22-13506-f006:**
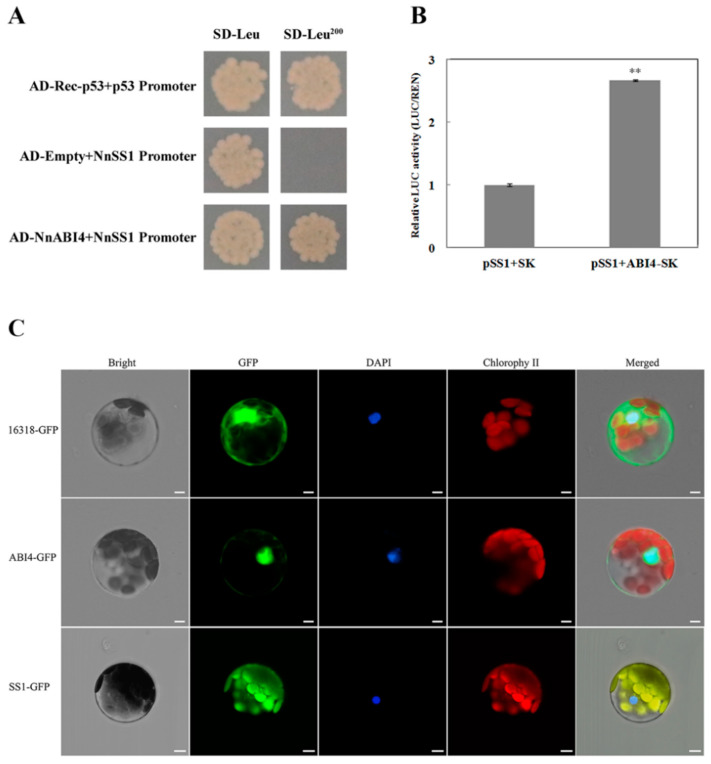
Activity analysis of NnAB14 promoting *NnSS1* and the subcellular localization of NnAB14 and NnSS1. (**A**) Y1H experiment showed that NnABI4 could directly bind to the promoter of *NnSS1*; (**B**) dual luciferase test verifies the activation of NnABI4 on the promoter of *NnSS1*. The ‘**’ above the histogram indicated the statistical significance at the level of 0.01 (*p* < 0.01). Error bars show SD from three biological replicates; (**C**) subcellular localization of NnABI4 and NnSS1 proteins. Bar = 5 um.

**Figure 7 ijms-22-13506-f007:**
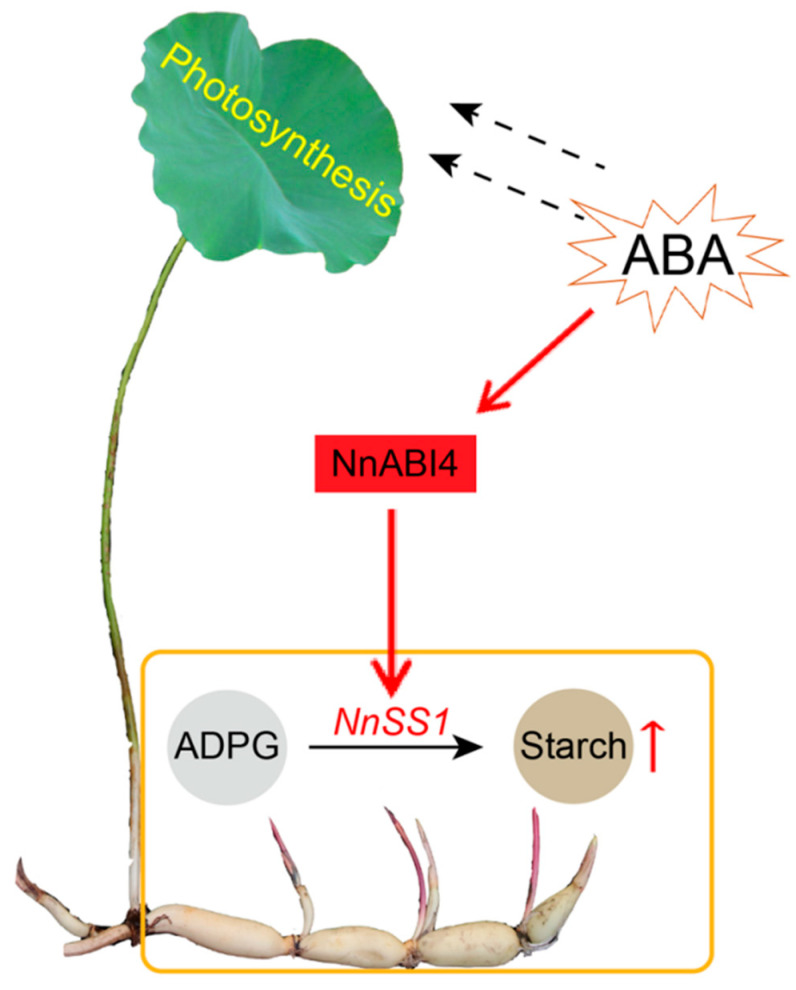
Pattern of ABA-regulating starch synthesis in lotus.

## Data Availability

Data are contained within the article or Appendix A.

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
