# Peer review of "NnABI4-Mediated ABA Regulation of Starch Biosynthesis in Lotus (*Nelumbo nucifera* Gaertn)"

_ijms, 2021, doi:10.3390/ijms222413506_

Round 1

Reviewer 1 Report

The present manuscript depicts the importance of ABA in regulating plant starch synthesis. The effect of exogenous ABA was studied on lotus leaves and photosynthetic characteristics were measured  and then qPCR expression analysis of starch synthesis-related genes and ABA pathway genes were determined. This study may provide scientific contribution of ABA regulation of starch synthesis network.

The discussion needs improvement to highlight own results and discuss them with those reported in the literature.

Conclusion is missing from the text.

Please improve the manuscript according to the following suggestions:

line 249: The main plant varieties... please provide the details of varieties used in present study. 

line 256: treatment groups...what are the treatment groups...?

line 257: Samples were taken at the beginning, early, mid- 257
dle and late stage of expansion of rhizome of lotus.....mention the days of sampling.

line 267: lotus rhizomes "are"......"were"

line 268: space between units " 1 g"

line 274: spacing error "qPCR(Vazyme,"..rectify it.

Reviewer 2 Report

The manuscript entitled “NnABI4-mediated ABA regulation of starch biosynthesis in lotus (Nelumbo nucifera Gaertn)” submitted by Peng Wu et al. described the effect of exogenous application of ABA on the lotus to investigate starch biosynthesis. The authors have evaluated the photosynthetic characteristics after ABA treatment. They also studied the expression profile of key genes implicated in the starch biosynthetic pathway after ABA treatment. After that, they performed yeast one-hybrid and dual luciferase assay to study whether NnABI4 protein can promote the expression of NnSS1 by directly binding to its promoter. Subcellular localization was performed to show that NnABI4 encodes a nuclear protein, and NnSS1 protein was located in the chloroplast. The present MS is very well written, clear, precise, and nicely represented. Though, the concept of the present study is good and informative. However, the manuscript needs minor revision. Please go through the following comments one by one carefully and correct them:

  • Page 1, line 29: please replace "maiz" with "maize".
  • Line 33: Please remove space after bracket in the sentence “synthase (GBSS),soluble”
  • Line 41: Line Please explain the abbreviations DP, when the first time it appears in the manuscript.
  • Section 2.1. lines 84-85: Please re-write the sentence “First, the stomatal conductance was higher than that of CK at 1 and 2 days after ABA treatment, while it was lower than CK at 3, 4, 5, and 6 days after ABA treatment”. Here please clarify stomatal conductance was higher…..in what? Also, check that text does not correlate with what the authors have presented in Figure 1A. Please recheck and correct.
  • Line 86-87: This sentence should be written as “The leaf transpiration rates were observed lower in ABA treated samples than CK after two to six days after ABA treatment".
  • Line 90: Please put -2 and -1 in superscript.
  • Line 95: Put the space in “(B)Transpiration”
  • Line 103: Please correct the sentence as “Higher in ABA treated than that of the control”
  • Line 114: Repeated gene name “NnSS3” should be deleted from the text.
  • Line 120: Put a space in “treatment(Figure 3, Table S4)”.
  • Line 123: Spacing is missing in “rhizomes(Figure 3, Table S4)”.
  • Line 154: A.thaliana, should be replaced with Arabidopsis thaliana as it was used first time in the MS.
  • Carefully check for spacing throughout the manuscript. A few examples are here “ Line 155 A.thaliana and B.rapa” line 159 “papaya and A.trichopoda” etc.
  • Line 181-185: names of the plasmids such as 35S:ABI4-GFP, 35S:SS1-GFP and 35S:GFP should be written in italics.
  • Line 214: Gene names “NnSS1, and NnSBE1” should be written in italics.
  • Line 267: In the sentence “The lotus rhizomes are dried fresh”. Replace “are” with “were”.
  • Lines 274-275: concentration of RNA that was used for cDNA synthesis should be mentioned.
  • Lines 277-279: Remove space in “µ L”. Should be written together “µL”. Please check and correct.
  • Line 281: name of the gene “ß-Actin” should be italic. Please provide the gene ID of ß-Actin.
  • Line 298: A. tumefaciens should be written in italic.
  • Line 299: please mention the composition of the infection solution.
  • Line 299: At what wavelength OD was measured. Mention it.
  • Line 302: Injected what Nicotiana tabacum or Nicotiana benthamiana? Please write.
  • Line 309: Arabidopsis thaliana should be italic
  • Line 311: What is the W5 solution? The author should mention the composition.
  • Line 314: How much protoplast was used for transformation purposes?
  • Line 315: What is the MMG solution?
  • What was the concentration of plasmid that was used for protoplast transformation?
